# MoGDE: Boosting Mobile Monocular 3D Object Detection with Ground Depth Estimation

**Yunsong Zhou** [1]  **Quan Liu** [1]  **Hongzi Zhu** [1*]  **Yunzhe Li** [1]  **Shan Chang** [2]  **Minyi Guo** [1]

[1]Shanghai Jiao Tong University   [2]Donghua University

{zhouyunsong,liuquan2017,hongzi,yunzhe.li,guo-my}@sjtu.edu.cn   changshan@dhu.edu.cn

## Abstract

Monocular 3D object detection (Mono3D) in mobile settings (*e.g.*, on a vehicle, a drone, or a robot) is an important yet challenging task. Due to the *near-far disparity* phenomenon of monocular vision and the ever-changing camera pose, it is hard to acquire high detection accuracy, especially for far objects. Inspired by the insight that the depth of an object can be well determined according to the depth of the ground where it stands, in this paper, we propose a novel Mono3D framework, called *MoGDE*, which constantly estimates the corresponding ground depth of an image and then utilizes the estimated ground depth information to guide Mono3D. To this end, we utilize a pose detection network to estimate the pose of the camera and then construct a feature map portraying pixel-level ground depth according to the 3D-to-2D perspective geometry. Moreover, to improve Mono3D with the estimated ground depth, we design an RGB-D feature fusion network based on the transformer structure, where the long-range self-attention mechanism is utilized to effectively identify ground-contacting points and pin the corresponding ground depth to the image feature map. We conduct extensive experiments on the real-world KITTI dataset. The results demonstrate that MoGDE can effectively improve the Mono3D accuracy and robustness for both near and far objects. MoGDE yields the best performance compared with the state-of-the-art methods by a large margin and is ranked number one on the KITTI 3D benchmark.

## 1  Introduction

Building on the promising progress achieved in 2D object detection in recent years [41, 40], 3D object detection, particularly on moving agents, has received increasing attention from both industry and academia as an important component in many applications, ranging from autonomous vehicles [17] and drones, to robotic manipulation and augmented reality applications. Compared to LiDAR-based [12, 36, 42, 43, 23, 63] and stereo-based [10, 11, 21, 35, 38, 54] methods, a much cheaper, more energy-efficient, and easier-to-deploy alternative, *i.e.*, monocular 3D object detection (Mono3D), remains an open and challenging research field. A practical Mono3D detector for moving agents should meet the following two requirements: 1) the 3D bounding box produced by the Mono3D detector should be accurate enough, not only for near objects but also for very distant objects, to secure, for instance, high-priority driving safety applications; 2) the Mono3D detector should remain robust in mobile scenarios, where the camera pose inevitably changes along with the movement of a mobile agent.

In the literature, recent Mono3D methods with complex network structures [62, 33, 52, 37, 48] have achieved considerably high accuracy for near objects, but the predicted 3D bounding boxes for far objects are often *ill-posed* due to the lack of depth cues. This huge disparity between near and far objects lies in the nature of monocular vision. Specifically, as depicted in Figure 1 (a), equal

---

*Corresponding author.

36th Conference on Neural Information Processing Systems (NeurIPS 2022).

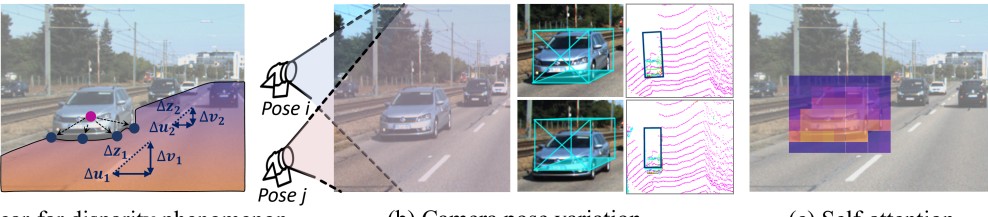

| (a) Near-far disparity phenomenon | (b) Camera pose variation | (c) Self-attention |

Figure 1: (a) Equal distances of different depths from the camera (*e.g.*, $\Delta z_1 = \Delta z_2$) have a distinct number of pixels in the image (*e.g.*, $\Delta u_1 > \Delta u_2$ and $\Delta v_1 > \Delta v_2$), which is referred to as the near-far disparity phenomenon of monocular vision, making the detection of far object susceptible to pixel rounding errors. (b) The camera pose variance, caused by the movement of a mobile agent, can eventually result in a large offset both in form of 3D boxes and in the bird's eye view. (c) Each color block represents its attention value with the centroid of the vehicle. The attention mechanism of the transformer network can be well leveraged for this long-range relationship modeling.

distances of different depth from the camera (*e.g.*, $\Delta z_1 = \Delta z_2$) have a distinct number of pixels in the image (*e.g.*, $\Delta u_1 > \Delta u_2$ and $\Delta v_1 > \Delta v_2$), which makes the pixel rounding errors have a non-negligible impact for detecting far objects. Furthermore, as illustrated in Figure 1(b), the camera pose variance can eventually result in a large offset both in form of 3D boxes and in the bird's eye view [15]. To the best of our knowledge, existing Mono3D methods such as geometric constraint based [9, 34, 6, 53], pseudo-LiDAR based [29, 30, 50, 54, 37, 16, 28], and pure image based [41, 45, 61, 3, 25, 49, 13, 22, 4, 62, 60, 27, 59], have not taken into account the issue of inevitable camera pose changes in mobile scenarios.

In this paper, we propose a novel Mono3D method, called *MoGDE*, which fixates on improving detection accuracy and robustness in mobile settings. We have one key insight that *the depth of an object in 3D space can be well determined according to the depth of the ground where it stands*. Given the pinhole model and the pose of a camera, the depth of each pixel corresponding to the ground can be accurately derived. Based on this insight, the core idea of MoGDE is first to constantly estimate the ground depth while moving and then to utilize the estimated ground depth information to guide a Mono3D detector.

There are two main challenges in designing MoGDE. First, how to detect varying camera pose (*e.g.*, the pitch and roll angles) from an image, which is dynamically changing in mobile scenes, and how to ultimately obtain accurate ground depth information are non-trivial. It is clear that different camera poses correspond to distinct ground depth estimates. To tackle this challenge, we introduce a pose detection network to extract the vanishing point and horizon information in an image to estimate the instant camera pose corresponding to this image. After the view direction of the camera is decided, we then construct a feature map portraying pixel-level depth clues. Specifically, we envision a virtual 3D scene containing only the sky and the ground and project this virtual scene to an image where each pixel is associated with a depth uniquely derived according to the 3D-to-2D perspective geometry. Therefore, MoGDE can obtain the dynamic ground depth information as prior knowledge for guiding Mono3D.

Second, how to incorporate the estimated ground depth into the image features to enhance the detection accuracy is challenging. Based on our aforementioned insight, it is essential for the Mono3D detector to identify those *ground-contacting points* of an object on the image. For example in Figure 1 (a), these blue dots denote the ground-contacting points of a vehicle. To this end, we design an RGB-D feature fusion network based on the transformer structure to tie the ground depth feature to the image feature. Specifically, as illustrated in Figure 1(c), the feature fusion network captures the feature of pixels close to the centroid of an object and identifies those ground-contacting points using the attention mechanism. It then attaches depth values with weights to compute a new feature map containing object location information. As a result, accurate 3D detection results can be obtained via a conventional Mono3D detector using the fused feature map.

Experiments on KITTI dataset [17] demonstrate that our method outperforms the SOTA methods by a large margin. Such a framework can be applied to existing detectors and is practical for industrial applications. The proposed MoGDE is ranked *number one* on the KITTI 3D benchmark by submission. The whole suite of the code base will be released and the experimental results will

be posted to the public leaderboard. We highlight the main contributions made in this paper as follows: 1) A novel Mono3D detector in a mobile setting is introduced, leveraging the dynamically estimated ground depth as prior knowledge to improve the detection accuracy and robustness for both near and far objects. 2) A transformer-based feature fusion network is designed, which utilizes the long-range attention mechanism to effectively identify ground-contacting points and pin the corresponding ground depth to the image feature map. 3) Extensive experiments on the real-world KITTI dataset are conducted and the results demonstrate the efficacy of MoGDE.

## 2  Related Work

**Monocular 3D Object Detection.** The monocular 3D object detection aims to predict 3D bounding boxes from a single given image. Existing Mono3D methods can be roughly divided into the following three categories. *1) Geometric constraint based methods:* Extra information of prior 3D vehicle shapes is widely used, such as vehicle computer aided design (CAD) models [9, 34, 6, 53, 26] or key points [1]. By this means, extra labeling cost is inevitably required. *2) Depth assist methods:* A standalone depth map of the monocular image is predicted at the first stage. Such prior knowledge can be derived in various ways, such as a depth map generated by LiDAR point cloud (or Pseudo-LiDAR) [50, 29, 8, 39], monocular depth predictors [37, 28, 16], or disparity map generated by stereo cameras [54]. However, such external data is not easily available in all scenarios. In addition, the inference time increases significantly due to the prediction of these dense heatmaps. *3) Pure image-based methods:* Without requiring extra side-channel information, such methods [20, 18, 24, 44] take only a single image as input and adopt center-based pipelines following conventional 2D detectors [41, 61, 45]. M3D-RPN [3] reformulates the monocular 3D detection problem as a standalone 3D region proposal network. With very few handcrafted modules, SMOKE [25] and FCOS3D [49] predict a 3D bounding box by combining a concise one-stage keypoint estimation with regressed 3D variables based on CenterNet [61] and FCOS [45], respectively. To further strengthen monocular detectors, current SOTA methods have introduced more effective but complicated geometric priors. MonoPair [13] improves the modeling of occluded objects by considering the relationship of paired samples and parses their spatial relations with uncertainty. Kinematic3D [4] proposes a novel method for monocular video-based 3D object detection, which uses kinematic motion to improve the accuracy of 3D localization. MonoEF [62] first proposes a novel method to capture the camera pose in order to formulate detectors that are not subject to camera extrinsic perturbations. MonoFlex [60] conducts an uncertainty-guided depth ensemble and categorizes different objects for distinctive processing. GUPNet [27] solves the error amplification problem by geometry-guided depth uncertainty and collocates a hierarchical learning strategy to reduce the training instability. MonoDETR [59] introduces a simple monocular object detection framework that makes the vanilla transformer to be depth-aware and enforces the whole detection process guided by depth. The above geometrically dependent designs largely promote the overall performance of center-based methods, but the underlying problem still exists, namely, the detection accuracy for distant objects is still not satisfactory.

**Object Detection with Transformer.** 2D object detectors [61, 45] have achieved excellent performance in recent years but are equipped with cumbersome post-processing, e.g. non-maximum suppression (NMS) [41]. To circumvent it, the pioneering work DETR [5] constructs a novel and simple framework by adapting the powerful transformer [47] to the field of vision detection. DETR detects objects on images by encoding-decoding paradigm, which improves the detection performance by using the long-range attention mechanism. DETR is further enhanced by designing deformable attention [64], placing anchors [51], setting conditional attention [31], embedding dense prior [57], and so on [32]. Some recent works have tried to apply transformer to some other tasks related to monocular scene reconstruction, depth prediction, *etc.* Transformerfusion [2] leverages the transformer architecture so that the network learns to focus on the most relevant image frames for each 3D location in the scene, supervised only by the scene reconstruction task. MT-SfMLearner [46] first demonstrates how to adapt vision transformers for self-supervised monocular depth estimation focusing on improving the robustness of natural corruptions. While these methods have made a demonstration of how to apply a transformer to a monocular camera model [14, 55, 56], they all rely on other branches (either the environment reconstruction or the depth map), which will not be available in a typical Mono3D task based on RGB images.

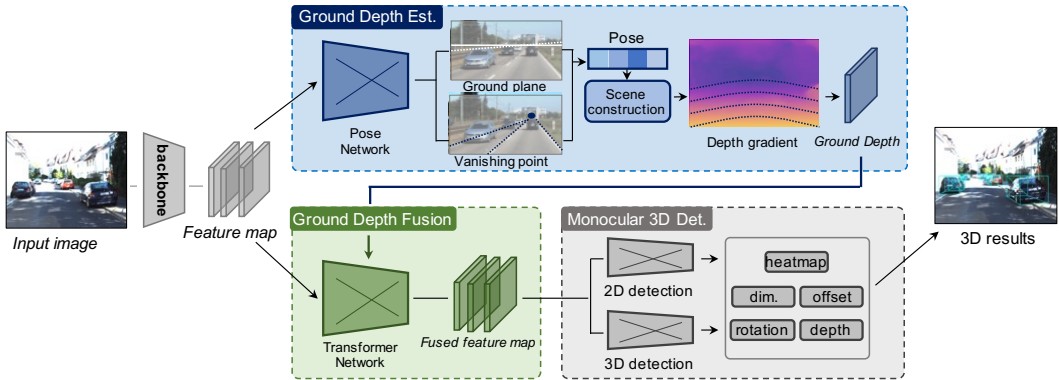

Figure 2: MoGDE consists of three main components, *i.e.*, *ground depth estimation* (GDE), *ground depth fusion* (GDF) and *monocular 3D detection* (M3D). In GDE, the pose network predicts the ground plane as well as the vanishing point. The derived pose information is then used to construct a virtual scene and obtain a pose-specific ground depth feature map. In GDF, a transformer network is leveraged to fuse the image features with the ground depth feature map, resulting a ground-aware fused feature map. M3D employs a standard Mono3D detector as the underlying detection core.

Our MoGDE inherits DETR's superiority for non-local encoding and long-range attention. Specifically, we endow the transformer to be *ground-aware* by pinning ground depth to image features leveraging the encoder-decoder architecture to improve the detection accuracy for far objects.

## 3 Design of MoGDE

### 3.1 Overview

The core idea of MoGDE is to utilize dynamically estimated ground depth information to improve Mono3D so that two goals can be achieved: 1) superior ground-aware image features are obtained to increase Mono3D accuracy for both near and far objects; 2) the impact of camera pose variation is diminished to enhance Mono3D robustness in mobile settings. Figure 2 depicts the architecture of our framework. Specifically, MoGDE first adopts the DLA-34 [58] as its backbone, which takes a monocular image of size $(W \times H \times 3)$ as input and outputs a feature map of size $(W_s \times H_s \times C)$ after down-sampling with an $s$-factor. Then, the feature map is fed into three components as follows:

**Ground Depth Estimation (GDE).** GDF mainly integrates two functions, *i.e.*, *camera pose detection* (CPD), and *virtual scene construction* (VSC). Specifically, CPD estimates the camera pose (*i.e.*, the pitch and roll angles) based on the predicted vanishing point and ground plane extracted by a pose detection network. VSC establishes a 2D ground depth feature map based on a pose-specific virtual 3D scene containing only the sky and the ground.

**Ground Depth Fusion (GDF).** GDF leverages the attention mechanism of a transformer network to fuse the image features with the ground depth feature map, resulting a superior ground-aware fused feature map.

**Monocular 3D Detection (M3D).** MoGDE employs GUPNet [27], a SOTA CenterNet [61] based SOTA monocular 3D object detector as its underlying detection core.

### 3.2 Ground Depth Estimation

#### 3.2.1 Camera Pose Detection

In order to generate a ground depth estimate, it is key to detect the camera pose given an image feature map. We have the following proposition:

**Proposition 1:** *Given a benchmark camera coordinate system* $\mathbf{P}^0$*, which is aligned with the ground plane coordinate systems, and the current camera coordinate system* $\mathbf{P}^i$*, which is not aligned with* $\mathbf{P}^0$ *due to camera movement, there exists a transformation matrix* $\mathbf{A}$ *between* $\mathbf{P}^i$ *and* $\mathbf{P}^0$ *that can be uniquely determined by pitch* $\theta_p$ *and roll* $\theta_p$ *angle changes of the camera.*)

Therefore, we introduce the subsequent neural network to learn the pitch $\theta_p$ and roll $\theta_p$ angle changes of the camera when the camera coordinate system changes from $\mathbf{P}^0$ to $\mathbf{P}^i$. Specifically, in addition to the regular regression tasks in CenterNet [61] based network, we introduce a regression branch for pose detection following MonoEF [62]. Since the camera pose is a feature that is implicit for images, we chose two physical quantities with a clear meaning for detection: the ground plane (associated with roll angle) and the vanishing point (associated with pitch angle). Following the state-of-the-art odometer framework in DeepVP [7], we represent a regression task with L1 loss as:

$$
\begin{aligned}
[\hat{\mathbf{y}}_{\text{gp}}, \hat{\mathbf{y}}_{\text{vp}}] &= f^{\text{pose}}\left(\mathbf{H}\right), \\
\mathcal{L}_{\text{pose}} &= \left\| \mathbf{A} - \mathbf{g}\left(\hat{\mathbf{y}}_{\text{gp}}, \hat{\mathbf{y}}_{\text{vp}}\right) \right\|,
\end{aligned}
\tag{1}
$$

where $\mathbf{H}$ is the input image feature; $f^{pose}$ is the CNN architecture used for horizon and vanishing point detection in the work [19]; $\hat{\mathbf{y}}_{\text{gp}}$ and $\hat{\mathbf{y}}_{\text{vp}}$ are the predicted ground plane and vanishing point; $\mathbf{g}$ is a mapping function $\mathbf{g} : \left(\mathbb{R}^2, \mathbb{R}^2\right) \mapsto \mathbf{A}_{3\times3}$ which turns pitch and roll angles into a matrix $\mathbf{A}$. The regression network is supervised by $\mathcal{L}_{\text{pose}}$ and can be trained jointly with other Mono3D branches.

### 3.2.2 Virtual Scene Construction

We envision such a virtual scene, where there is a vast and infinite horizontal plane in the camera coordinate system $\mathbf{P}^0$, and have the following proposition:

**Proposistion 2:** *Given the camera coordinate system $\mathbf{P}^i$, the virtual horizontal plane can be projected on the image plane of the camera according to the ideal pinhole camera model and the depth corresponding to each pixel on the image is determined by the camera intrinsic parameter $\mathbf{K}$ and pose matrix $\mathbf{A}$ from $\mathbf{P}^0$ to $\mathbf{P}^i$.*

We first construct the ground depth feature map in the camera coordinate system $\mathbf{P}^0$. Specifically, as illustrated in Figure 3, for each pixel on the depth image locating at $(u^0, v^0)$ with an estimated depth $\hat{z}^0$, it can be back-projected to a point $(x_{3d}^0, y_{3d}^0, \hat{z}^0)$ in the 3D scene:

$$
x_{3d}^0 = \frac{u^0 - c_x}{f_x}\hat{z}^0 \quad y_{3d}^0 = \frac{v^0 - c_y}{f_y}\hat{z}^0,
\tag{2}
$$

where $f_x$ and $f_y$ are the focal lengths represented in the units of pixels along the $x$- and $y$-axis of the image plane and $c_x$ and $c_y$ are the possible displacement between the image center and the foot point. These are referred to as the camera intrinsic parameters $\mathbf{K}$. We omit the camera extrinsic $\mathbf{T}$ for the sake of simplicity, and the depth corresponding to each pixel on the image is solely determined by the camera intrinsic parameter $\mathbf{K}$ under $\mathbf{P}^0$.

Assume that the elevation of the camera from the ground, denoted as $EL$, is known (for instance, the mean height of all vehicles in the KITTI dataset, including ego vehicles, is 1.65m [17]), the depth of a point on the depth feature map $(u^0, v^0)$ can be calculated as:

$$
z^0 = \frac{f_y \cdot EL}{v^0 - c_y}.
\tag{3}
$$

Note that (3) is not continuous when the point is near the vanishing point, *i.e.*, $v^0 = c_y$, and does not physically hold when $v^0 \leq c_y$. To address this problem, similar to the KITTI stereo setup, we encode the depth gradient value as an associated feature map using a virtual stereo setup with baseline $B = 0.54$m. We represent the ground depth $d$ in the following form:

$$
d = \text{ReLU}(f_y \cdot B \frac{v^0 - c_y}{f_y \cdot EL + b})
\tag{4}
$$

where $b$ is a constant to prevent the value of $d$ from being too large. The ReLU activation is applied to suppress ground depth values smaller than zero, which is not physically feasible for monocular cameras. As a result, the ground depth feature map becomes spatially continuous and consistent.

Finally, to obtain the ground depth feature map in $\mathbf{P}^i$, the model needs to convert the 3D coordinate system first, and then just apply (4). We omit the formula derivation due to page limitation.

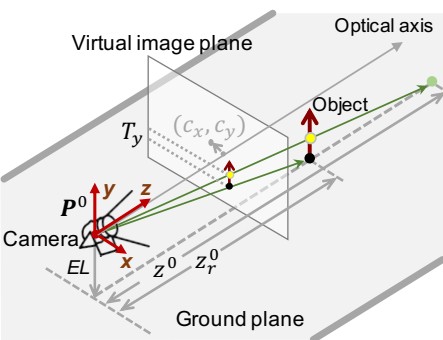

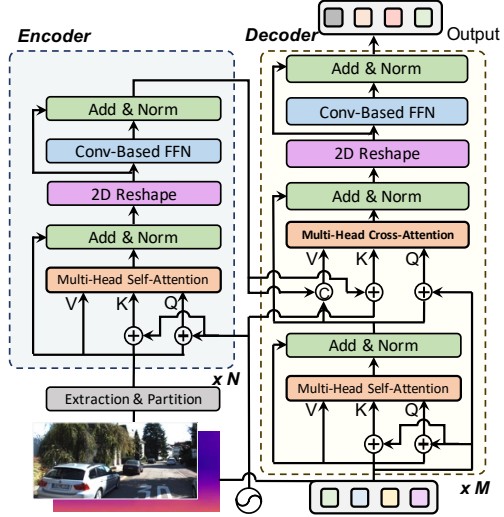

Figure 3: Perspective geometry for ground depth estimation. In the camera coordinate system $\mathbf{P}^0$, given the camera intrinsic parameters $\mathbf{K}$ and the elevation of the camera from the ground $EL$, the depth of a point on the ground depth feature map can be calculated. Moreover, by estimating the transformation matrix $\mathbf{A}$ between $\mathbf{P}^0$ and an arbitrary $\mathbf{P}^i$, the ground depth feature map in $\mathbf{P}^i$ can be obtained. To utilize the ground depth feature, it is key to locate *ground-contacting* points of an object (*e.g.*, the dark point) to get an accurate depth (*e.g.*, $z_r^0$). On the contrary, misuse of the depth in the ground depth feature corresponding to other points on the object (*e.g.*, the bright point) leads to obvious depth estimation error (*e.g.*, $z^0$).

Figure 4: The architecture of *ground-aware* transformer. The encoder uses self-attention to encode the non-local mutual correlation of image pixels (*i.e.*, object centers and ground points). The ground depth estimate is used to generate location queries which are thereby fed to the decoder along with the location encoding. The cross-attention in the decoder prompts each query to consider the image and depth features of its associated points.

## 3.3 Ground Depth Fusion

In real-world scenarios, as depicted in Figure 3, objects have height. To fuse the image feature and the ground depth feature, it is key to locate *ground-contacting* points of an object (*e.g.*, the dark point) to get an accurate depth (*e.g.*, $z_r^0$). On the contrary, misuse of the depth in the ground depth feature corresponding to other points on the object (*e.g.*, the bright point) leads to obvious depth estimation error (*e.g.*, $z^0$). Specifically, the relation between the estimated depth of an object $\hat{z}^0$ and the pixel displacement in locating ground-contacting points, denoted as $T_y$, can be calculated as,

$$\hat{z}^0 = \frac{EL \cdot f_y \cdot z_r^0}{EL \cdot f_y - z_r^0 \cdot T_y}. \tag{5}$$

It can be seen from (5) that $T_y$ can cause inaccurate $\hat{z}^0$. However, how to acquire $T_y$ is non-trivial. Inspired by the great success of transformer [5, 64, 51, 31, 57, 32] in adaptive long-range relational modeling, we propose a *ground-aware* feature fusion method based on a transformer structure as depicted in Figure 4, leveraging its attention mechanism to automatically locate ground-contacting points of an object and fuse the corresponding depth feature with the image feature of that object.

**Encoder.** Our transformer encoder aims to encode the correlation between image features using a self-attention mechanism. The input of the transformer encoder is the flattened image features $\mathbf{H}_{\text{img}} \in \mathbb{R}^{N \times C}$ with position encoding and the output is the embedding vectors $\mathbf{H}_e \in \mathbb{R}^{N \times C}$ to be sent to the decoder. Following the self-attention pipeline, given the input matrix calculated from the image features: query $\mathbf{Q} \in \mathbb{R}^{N \times C}$, key $\mathbf{K} \in \mathbb{R}^{N \times C}$, and value $\mathbf{V} \in \mathbb{R}^{N \times C}$ with sequence length $N = W \times H$, the output of $l + 1$-th layer of self-attention can be briefly formulated as:

$$\mathbf{Q}^l, \mathbf{K}^l, \mathbf{V}^l = \text{Embedding}(\mathbf{H}_{\text{img}}^l, \mathbf{W}_q^l, \mathbf{W}_k^l, \mathbf{W}_v^l),$$
$$\mathbf{H}_{\text{img}}^{l+1} = \text{Attention}(\mathbf{Q}^l, \mathbf{K}^l, \mathbf{V}^l) = \text{softmax}\left(\mathbf{Q}^l \mathbf{K}^{l\top} / \sqrt{C}\right) \mathbf{M}^l \mathbf{V}^l. \tag{6}$$

Here, $\mathbf{M}^l$ is the mask used to constrain the visible range of attention. The introduction of $\mathbf{M}^l$ is to take advantage of the *ground-aware* property (*i.e.*, the depth of each object should be related to the

depth of the object's location) so that each pixel will only consider information within a window around that location. The encoded feature obtained through multi-head self-attention operation is then re-transformed into a 2D feature map format and fed into a convolution-based feed-forward network (FFN). The 2D reshape as well as convolution-based FFN are necessary because image data is two-dimensional, unlike one-dimensional serialized data.

**Decoder.** The proposed transformer decoder aims to determine for each location its depth information, using the cross-correlation between the ground depth and the image features. We propose utilizing the ground depth as the location query of the decoder instead of learnable embedding (object query), which is different from the common usage in previous encoder-decoder vision transformer works [5, 64]. The main reason is that the simple learnable embedding is hard to fully represent the object's property and handle complex depth variant situations in the Mono3D task. In contrast, plentiful distance-aware cues are hidden in the ground depth features, which will give the transformer a baseline estimate of the expected depth at each location. To this end, the decoder can leverage the power of cross-attention in the transformer to efficiently model the correlation between the target pixel point and the point of interest (*i.e.*, the grounded point), thus achieving the *ground-awareness* for higher performance.

Specifically, the input of the decoder is the flattened ground depth $\mathbf{H}_{dep} \in \mathbb{R}^{N \times 1}$ with position encoding and embedding vectors $\mathbf{H}_e \in \mathbb{R}^{N \times C}$ obtained from the encoder. In addition, the output is the aggregated feature map $\mathbf{H}_d \in \mathbb{R}^{N \times C}$. The ground depth is first embedded upon the standard self-attention architecture following (6). For the cross-attention module, its input $\mathbf{Q}$ is derived from the self-attention part upstream in the decoder, and its $\mathbf{K}$ is derived from the encoder. The input $\mathbf{V}$ is a concatenation of two sources from both the encoder and decoder. The purpose of this concatenation is to make the decoder take into account both the information from the image and the depth during decoding.

## 4 Performance Evaluation

We conduct experiments on the widely-adopted KITTI3D dataset and KITTI Odometry dataset [17]. We report the detection results with three-level difficulties, *i.e.* easy, moderate, and hard, in which the moderate scores are normally for ranking and the hard category is generally distant objects that is difficult to distinguish.

### 4.1 Quantitative and Qualitative Results

We first show the performance of our proposed MoGDE on KITTI 3D object detection benchmark [2] for car. Comparison results with other state-of-the-art (SOTA) monocular 3D detectors are shown in Table 1. For the official *test* set, we achieve the highest score for all kinds of samples and are ranked

| Method | Extra data | Test, $AP_{3D}$ | | | Test, $AP_{BEV}$ | | |
|---|---|---|---|---|---|---|---|
| | | Easy | Mod. | Hard | Easy | Mod. | Hard |
| PatchNet [28] | Depth | 15.68 | 11.12 | 10.17 | 22.97 | 16.86 | 14.97 |
| D4LCN [16] | Depth | 16.65 | 11.72 | 9.51 | 22.51 | 16.02 | 12.55 |
| Kinematic3D [4] | Multi-frames | 19.07 | 12.72 | 9.17 | 26.69 | 17.52 | 13.10 |
| MonoRUn [8] | Lidar | 19.65 | 12.30 | 10.58 | 27.94 | 17.34 | 15.24 |
| CaDDN [39] | Lidar | 19.17 | 13.41 | 11.46 | 27.94 | 18.91 | 17.19 |
| AutoShape [26] | CAD | 22.47 | 14.17 | 11.36 | 30.66 | 20.08 | 15.59 |
| SMOKE [25] | None | 14.03 | 9.76 | 7.84 | 20.83 | 14.49 | 12.75 |
| MonoFlex [60] | None | 19.94 | 13.89 | 12.07 | 28.23 | 19.75 | 16.89 |
| GUPNet [27] | None | 20.11 | 14.20 | 11.77 | - | - | - |
| MonoDETR [59] | None | 23.65 | 15.92 | 12.99 | 32.08 | 21.44 | 17.85 |
| **MoGDE (Ours)** | | **27.07** | **17.88** | **15.66** | **38.38** | **25.60** | **22.91** |
| *Improvement* | *v.s. second-best* | +3.42 | +1.96 | +2.67 | +6.30 | +4.16 | +5.06 |

Table 1: $AP_{40}$ scores(%) of the car category on KITTI *test* set at 0.7 IoU threshold referred from the KITTI benchmark website. We utilize bold to highlight the best results, and color the second-best ones and our performance gain over them in blue. Our model is ranked NO. 1 on the benchmark.

---

[2]http://www.cvlibs.net/datasets/kitti/eval object.php?obj benchmark=3d

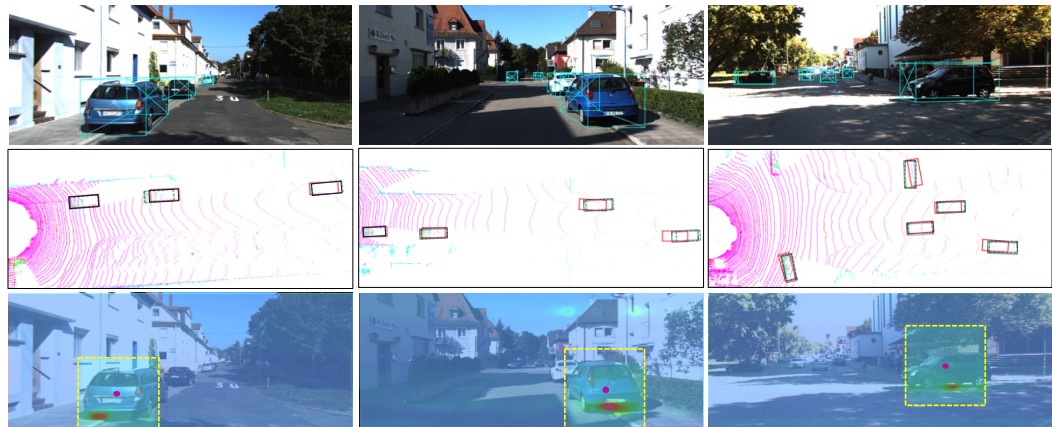

Figure 5: Qualitative results on KITTI Odometry dataset. The predicted 3D bounding boxes of our proposed MoGDE are shown in the first row. The second row shows the detection results in the bird's eye view ($z$-direction from right to left). The green dashed boxes are the ground truth, and the blue and red solid boxes are the prediction results of our MoGDE and the comparison baseline (GUPNet [27]), respectively. The third row visualizes the results of the attention map in the transformer's encoder, where the purple point is the location of the query point; the yellow dashed box is the range of the encoder's mask; the brightness of the image represents the attention value between the query point and that pixel.

No.1 among all existing methods with different additional data inputs on all metrics. Compared to the second-best models, MoGDE surpasses them under easy, moderate, and hard levels respectively by +3.42, +1.96, and +2.67 in $AP_{3D}$, especially achieving a significant increase (17%) in the hard level. The comparison fully proves the effectiveness of the proposed oracle fusion for images with prior depth knowledge.

Figure 5 shows the qualitative results on the KITTI Odometry dataset. Compared with the baseline model without the aid of ground depth, the predictions from MoGDE are much closer to the ground truth, especially for distinct objects. It shows that the consideration of sight-based supporting depth clues can help to locate the object precisely.

## 4.2 Ablation Study

**Effectiveness of each proposed component.** In Table 2, we conduct an ablation study to analyze the effectiveness of the proposed components: (a) Baseline: only using image features for 3D object detection, *i.e.*, without concerning posed variance and proposed ground-aware modules. (b) Considering the camera pose variations implied in the images, we use the method described in [62] to apply a "projection transform" to the input image to remove the perturbations. (c) Considering the use of ground plane clues, we generate a depth oracle about the scene (assuming constant pose) using a convolutional neural network. (d) With

|     | Pose -guided | Conv. Fusion | Tran. Fusion | Easy | Mod. | Hard |
|-----|:---:|:---:|:---:|:---:|:---:|:---:|
| (a) | - | - | - | 22.76 | 16.46 | 13.72 |
| (b) | ✓ | - | - | 22.78 | 16.93 | 14.04 |
| (c) | - | ✓ | - | 22.82 | 17.22 | 14.51 |
| (d) | - | - | ✓ | 22.93 | 18.42 | 15.46 |
| (e) | ✓ | ✓ | - | 23.07 | 18.66 | 15.73 |
| (f) | ✓ | - | ✓ | **23.35** | **20.35** | **17.71** |

Table 2: Effectiveness of different components of our approach on the KITTI *val* set for car category. The first column is whether the model takes into account the pose variance. The second and third columns show which way the model chooses to fuse the ground depth information.

the proposed *ground-aware* transformer, this model has the ability to model long-range relationships of pixels. (e) Full model except that we use a convolutional neural network for oracle fusion. (f) Full model (MoGDE).

First, we can observe from (a → b, c → e, and d → f) that there is an implicit uncalibrated pose variation in the KITTI dataset, and considering it is necessary to improve the detection accuracy. Besides, by observing (b → e), we illustrate that leveraging ground depth brings an improvement in accuracy in hard level, but the improvement is limited because fusion by convolution is clumsy.

| Pose Var. | Method | Val, $AP_{3D}$ | | | Val, $AP_{BEV}$ | | |
|---|---|---|---|---|---|---|---|
| | | Easy | Mod. | Hard | Easy | Mod. | Hard |
| Tiny | w/o | 20.64 | 14.87 | 12.47 | 27.97 | 20.78 | 17.79 |
| | w/ | **22.30** | **19.42** | **16.84** | **29.86** | **24.28** | **23.15** |
| | *Imp.* | +1.66 | +4.55 | +4.37 | +1.89 | +3.50 | +5.35 |
| Medium | w/o | 17.76 | 12.98 | 10.78 | 24.43 | 18.06 | 15.46 |
| | w/ | **21.86** | **19.08** | **16.61** | **29.23** | **23.70** | **22.84** |
| | *Imp.* | +4.10 | +6.10 | +5.83 | +4.81 | +5.64 | +7.38 |
| Large | w/o | 13.29 | 9.60 | 8.05 | 18.05 | 13.34 | 11.57 |
| | w/ | **21.10** | **18.35** | **16.10** | **28.33** | **22.98** | **22.06** |
| | *Imp.* | +7.81 | +8.75 | +8.05 | +10.28 | +9.64 | +10.48 |

Table 3: Robustness test of our approach on the KITTI *val* set for car category. Tiny, medium, and large correspond to three different degrees of posture change, *i.e.*, the camera pitch and roll angles vary with a Gaussian distribution with mean 0 and standard deviation 1, 2, and 3, respectively.

| Method | Val, $AP_{3D}$ | | | Val, $AP_{BEV}$ | | |
|---|---|---|---|---|---|---|
| | Easy | Mod. | Hard | Easy | Mod. | Hard |
| M3D-RPN [3] | 14.53 | 11.07 | 8.65 | 20.85 | 15.62 | 11.88 |
| M3D-RPN + **Ours** | **19.85** | **16.84** | **14.62** | **25.16** | **20.65** | **17.39** |
| *Imp.* | +5.32 | +5.77 | +5.97 | +4.31 | +5.03 | +5.51 |
| MonoPair [13] | 16.28 | 12.30 | 10.42 | 24.12 | 18.17 | 15.76 |
| MonoPair + **Ours** | **19.20** | **15.42** | **13.16** | **27.33** | **21.71** | **18.68** |
| *Imp.* | +2.92 | +3.12 | +2.74 | +3.21 | +3.54 | +2.92 |
| Kinematic3D [4] | 19.76 | 14.10 | 10.47 | 27.83 | 19.72 | 15.10 |
| Kinematic3D + **Ours** | **21.59** | **16.54** | **12.77** | **29.80** | **22.80** | **17.96** |
| *Imp.* | +1.83 | +2.44 | +2.30 | +1.97 | +3.08 | +2.86 |

Table 4: Extension of MoGDE to existing image-only monocular 3D object detectors. We show the $AP_{40}$ scores(%) evaluated on KITTI3D *val* set. **+Ours** indicates that we apply the GDE and GDF modules to the original methods. All models benefit from the MoGDE design.

In contrast, (e → f) indicates the effectiveness of the transformer, which helps the model to understand the long-range attention relationship between pixel points and the ground plane.

**Visualization of attention.** To facilitate the understanding of our *ground-aware* transformer, we visualize the depth self-attention map in the encoder and paint the query points red and mask region yellow in the third row of Figure 5. As shown in the figure, it can be seen that, within the relevant region of each query, areas that are interfacing the object and the ground have the highest attention scores. In contrast, for non-ground pixels of the object, the lower attention values indicate that the query is not relevant to them, even if they have similar image features and are geographically adjacent. This implies that under the transformer's attention mechanism guidance, the query is able to borrow depth information from regions of interest (*i.e.*, the ground plane), which helps the fused feature map produce more accurate prediction results.

**Simulation experiments on robustness.** In order to verify the robustness of our proposed MoGDE against camera pose variance, we set three cases of variances (tiny, medium, and large) to compare the accuracy degradation of MoGDE with that of the baseline. In Table 3, it can be noticed that the baseline is quite sensitive to pose variance, with very severe performance degradation, while our model only has a slight performance drop. Moreover, our model performs more robustly especially in the hard case, gaining higher performance improvement. This demonstrates the effectiveness of our proposed pose-specific ground depth in handling camera pose variance for mobile scenes.

**Plugging into existing methods.** Our proposed approach is flexible to extend to existing image-only Mono3D detectors. We respectively plug the Ground Depth Estimation and the Ground Depth Fusion components to three popular Mono3D detectors, which is shown in Table 4. It can be seen that, with the aid of our proposed ground depth fusion, these detectors can achieve further improvements on KITTI3D *val* set, demonstrating the effectiveness and flexibility of our approach. Particularly, MoGDE-enabled models tend to achieve more performance gains on the hard category. For example, for Kinematic3D, the $AP_{3D}/AP_{BEV}$ gain is +1.83/+1.97 on the easy category and +2.30/+2.86 on the hard category.

## 5 Conclusion

In this paper, we have proposed a Mono3D framework, called *MoGDE*, which can effectively utilize the estimated ground depth as prior knowledge to improve Mono3D in mobile settings. The advantages of MoGDE are two-fold: 1) it can significantly improve the Mono3D accuracy, especially for far objects, which is an open issue for Mono3D; 2) it can improve the robustness of Mono3D detectors when applied in more appealing mobile applications. Nevertheless, MoGDE still has two main limitations as follows: 1) it heavily relies on pose detection, which directly affects the accuracy of the ground depth estimation; 2) it also counts on the detection of ground-contacting points. In cases when such points are uncertain or ambiguous due to occlusion and truncation, it is hard for the proposed ground-aware feature fusion method to obtain accurate results. These limitations also direct our future work. We have implemented Mono3D and conducted extensive experiments on the real-world KITTI dataset. MoGDE yields the best performance compared with the state-of-the-art methods by a large margin and is ranked number one on the KITTI 3D benchmark.

## Acknowledgements

This research was supported in part by National Key RD Program of China (Grants No. 2018YFC1900700), National Natural Science Foundation of China (Grants No. 61872240 and 61972081), and the Natural Science Foundation of Shanghai (Grant No. 22ZR1400200).

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
