# OpenReview forum: "MoGDE: Boosting Mobile Monocular 3D Object Detection with Ground Depth Estimation"
_NeurIPS.cc/2022/Conference — NeurIPS 2022 Accept_

### Official Review · Reviewer_EteJ · 2022-07-10

**Rating:** 5
**Confidence:** 4
**Soundness:** 2 fair
**Presentation:** 2 fair
**Contribution:** 3 good

**Summary:**

This paper proposes a framework, namely MoGDE, to consider ground depth cues for more accurate and robust monocular 3D object detection. Specifically, it estimates the camera pose at first and constructs ground depth map to guide the depth estimation of objects. The fusion of ground depth and 2D features is achieved with a depth-aware DETR formulation. Finally, the overall framework achieves state-of-the-art results on the KITTI benchmark, and the qualitative results support the claim that the depth information of the ground assists the localization of 3D objects.

**Questions:**

See the weaknesses.

**Limitations:**

The author adequately analyzes the limitation of the proposed methods in the Conclusion Section.

**Strengths And Weaknesses:**

Strengths:
- The paper has a clear motivation, i.e., to use ground depth cues to assist the depth estimation of 3D objects.
- The basic idea is easy to follow, indicating that the presentation of this paper is overall smooth.
- The methodology part is overall clear and rigorous, although there are some points that require clarification.
- The performance on the KITTI test set is impressive, and the qualitative results well support the core idea of methodology.

Weaknesses:
- There are some confusing points in the methodology part:

    a. line 143: The camera pose typically consists of 6 DoF? including 3-DoF translation and 3-DoF rotation. Nevertheless, the author directly takes it as pitch and roll angles, which can be inaccurate and confusing especially when using this concept for the first time.

    b. line 170: How do the "predicted ground plane and vanishing point" formulated? It seems that there is no explanation for their specific definition.

    c. line 171: The matrix A not need any extra annotations? May need to clarify how the matrix is involved in the training procedure, at least more clearly in the main paper.

    d. line 182: The projection matrix in KITTI involves several offset factors, such as bx, by, bz to describe the baseline offset between camera 2 and the referenced camera. How do you deal with such details when deriving the projection equations?

    e. line 193: Why can the virtual stereo setup address the problem that the point near the vanishing point is not continuous? Here needs many words to clarify.

- Experiments:

    a. There seems no ablation or detailed results to reveal that the proposed method has boosted the detection performance for far-away objects, but the author takes this point as an important contribution in the presentation.

    b. Why need experiments on the Odometry Benchmark and why only show the qualitative results on that? Is there any difference between 3D detection Benchmark and Odometry Benchmark? How about the quantitative results on the Odometry Benchmark?

    c. line 267: lack of the explanation for the experiment group (d) (seems there is a typo)

    d. Why does the proposed method perform a little worse on the Easy split of the validation set?

    e. Would be better and more convincing to have experiments on a second dataset, such as nuScenes or Waymo.

- Minor comments:

    a. Can further polish the paper and fix some minor typos, such as need -> needs in line 198.

    b. Related works can be included:
        Ground-aware Monocular 3D Object Detection for Autonomous Driving, RA-L 2021.
        Probabilistic and Geometric Depth: Detecting Objects in Perspective, CoRL 2021.

---

> ### Author Response · Authors · 2022-08-02
> **Response to Reviewer EteJ**
>
> ## W1
> ***a.***
> When the camera is mounted, its position is fixed rather than changing at any time, so the 3-Dof translation of the camera is equivalent to none. The change of the camera's yaw angle does not change the vanishing point and the horizon, so it does not need to be considered for the ground depth estimation.
>
> ***b.***
> The horizon line is the baseline that determines the perspective of objects in the picture and the vanishing point is the point where parallel road boundaries converge on the horizon. They are only intermediate representations of the results of the pose detection network and can be defined formally as the line $a \cdot u+b \cdot v +c =0$ and the point $(u_{vp}, v_{vp})$.
>
> ***c.***
> Our pose estimation network follows the framework of DeepVP[7]. In Eq (1), the matrix $ \mathbf{A}$, formulated in Eq(9), represents the transfer matrix of the camera related to the motion of ego-vehicle (pitch and roll angles). During training, horizons and vanishing points are only used as intermediate results; they are not supervised with ground truth labels for training. The training label is the ego-vehicle motion transfer matrix after considering only pitch and roll angles (corresponding to matrix $\mathbf{A}$) provided by the KITTI Odometry dataset.
>
> ***d.***
> Only one camera is used in our model, so there is no need to consider the coordinate conversion between different cameras. The only thing to note is that the camera coordinate system of the color camera (camera #2) needs to be multiplied by the transfer matrix to camera #0 when projecting to the image coordinate system.
>
> ***e.***
> For an autopilot scene, basically only the depth of the part below the horizon is meaningful, the depth of the part above the horizon (usually the background and the sky) is infinite. To have an applicable measure of the depth estimation error, the KITTI Stereo dataset setting uses the inverse of depth values to circumvent the effect of infinity values. We follow their example and also use the inverse to circumvent the infinity values and make the values continuous.
>
> ## W2
> ***a.***
>
> |     Method    |       | Val,$AP_{3D}$ |       |       | Val,$AP_{BEV}$ |       |
> |:-------------:|:-----:|:-------------:|:-----:|:-----:|:--------------:|:-----:|
> |               |  Near |     Middle    |  Far  |  Near |     Middle     |  Far  |
> |     GUPNet    | 48.44 |     17.80     |  2.12 | 61.21 |      25.01     |  3.24 |
> |     MoGDE     | 49.64 |     22.45     | 10.94 | 63.17 |      29.74     | 11.62 |
> | _Improvement_ | +1.20 |     +4.65     | +8.82 | +1.96 |      +4.73     | +8.38 |
>
> In this table, we present the detection accuracy (%) of MoGDE with GUPNet as the baseline in the KITTI \textit{val} set for three distance ranges: near (5m-10m), middle (25-30m), and far (>50m). From this, we can see that our method has a significant improvement in the detection accuracy of distant objects.
>
> ***b.***
> At the dataset level, the image information of KITTI 3D and KITTI Odometry is basically the same, differing only in that the images of Odometry are continuous time series of data. We just think the Odometry dataset contains more vehicle targets and is more suitable to visualize the results of the vehicle's bbox. However, from the perspective of labeled ground truth, the 3D dataset provides the 3D bbox of the target in the scene, while the Odometry dataset provides the motion information between two adjacent frames. Our model is not specialized in visual odometry tasks so we do not present the results on its benchmark.
>
> ***c.***
> We make a typo here, we incorrectly write (d) for (c). Thanks for pointing out this, and we will correct it in the revision.
>
> ***d.***
> The first possible reason is the data imbalance of different splits during training. Another possible reason is that the other models basically focus on improving the overall accuracy, while our model focuses on improving the accuracy of the hard split. This leads to the fact that our model does not match the accuracy improvement of other models on the easy split.
>
> ***e.***
> The 3D object det. leaderboards of the Waymo and nuScenes datasets are evaluated and ranked using six cameras around the vehicle. The system model of our model is built exclusively on the front camera, so we did not experiment on these two datasets to ensure fairness of the comparison. In the revision, we will perform validation experiments using a front view camera on the Waymo and nuScenes datasets to illustrate the generalization ability of MoGDE on other datasets.
>
>
> ## W3
> ***a.***
> Thanks, we will correct them in the revision.
>
> ***b.***
> We will add these citations to the revision. The former introduces a ground-aware convolution module for Mono3D, providing geometric priors for the network to reason based on ground plane depth. The latter leverages the geometric relationship in perspective to construct a graph connecting instance estimations with uncertainty and thus predicts depths more accurately.

---

### Official Review · Reviewer_wuFE · 2022-07-11

**Rating:** 5
**Confidence:** 5
**Ethics Flag:** Yes
**Soundness:** 4 excellent
**Presentation:** 3 good
**Contribution:** 3 good

**Summary:**

This paper utilizes ground depth as geometry cue to improve monocular 3D detection. The ground depth is estimated by camera pose and vanishing point. And the ground depth information is further integrated into the network by the transformer. The experiments demonstrate the effectiveness of the proposed components. And it achieves state-of-art results on the KITTI benchmark.

**Questions:**

1. The paper mentions the Mono3D performance of far objects many times. How about the accuracy of objects at long distances (>50m)?

2. The experiments are only performed on the KITTI dataset. Is this method evaluated on other public datasets (e.g. Waymo, Nuscenes)?

**Ethics Review Area:**

["I don’t know"]

**Limitations:**

- As mentioned in the conclusion, the ground depth heavily relies on pose estimation. And it is difficult to ensure the accuracy of pose estimation.

- The method assumes that the ground is flat. However, the ground at the long distance and at the side is inconsistent with the nearby place. Therefore the virtual scene constructed from the infinite plane is not convinced.

- The formulas adopted in the equations are not unified. For example, the EL is suggested to be replaced by a single letter.


**Strengths And Weaknesses:**

+ The motivation is good. The ground depth is more reliable in the Mono3D task than other geometry clues.

+ The fusion of ground depth is well-structured, in which the ground-aware transformer locates the ground-contacting points and integrates their features.

+ The method is easy to follow. It is flexible to extend to other Mono3D models.

+ Extensive experiments show the effectiveness of the proposed components. And the method achieves SOTA results on the KITTI leaderboard.

---

> ### Author Response · Authors · 2022-08-02
> **Response to Reviewer wuFE**
>
> ***Q1: The paper mentions the Mono3D performance of far objects many times. How about the accuracy of objects at long distances (>50m)?***
>
> |        Method        |  |    Val,$AP_{3D}$    |       |  |    Val,$AP_{BEV}$    |       |
> |:--------------------:|:-------------:|:------:|:-----:|:--------------:|:------:|:-----:|
> |                      | Near          | Middle | Far   | Near           | Middle | Far   |
> | GUPNet               |         48.44 | 17.80  |  2.12 |          61.21 |  25.01 |  3.24 |
> | MoGDE                |         49.64 |  22.45 | 10.94 |          63.17 |  29.74 | 11.62 |
> | *Improvement*|         +1.20 |  +4.65 | +8.82 |          +1.96 |  +4.73 | +8.38 |
>
> In this table, we present the detection accuracy of MoGDE in the KITTI *val* set for three distance ranges: near (5m-10m), middle (25-30m), and far (>50m) with GUPNet as the baseline. From this, we can see that our method has a significant improvement in the detection accuracy of distant objects.
>
> ***Q2: The experiments are only performed on the KITTI dataset. Is this method evaluated on other public datasets (e.g. Waymo, Nuscenes)?***
>
> The 3D object detection leaderboards of the Waymo and nuScenes datasets are evaluated and ranked using six cameras around the vehicle. The system model of our proposed MoGDE is built exclusively on the front camera, so we did not experiment on these two datasets to ensure fairness of the comparison. In the revision, we will perform validation experiments using a front view camera on the Waymo and nuScenes datasets to illustrate the generalization ability of MoGDE on other datasets.
>
> ***L1: As mentioned in the conclusion, the ground depth heavily relies on pose estimation. And it is difficult to ensure the accuracy of pose estimation.***
>
> In L308 and Appendix C, we explain two main limitations of MoGDE. First, the model heavily relies on the detection of the ground-contacting points. When these points are uncertain or invisible due to occlusion and truncation, our proposed ground-aware feature fusion method does not yield better results. To illustrate this limitation by visualization, we give the bad cases due to truncation in Figure 10 in the supplementary materials. In our future work, to tackle this limitation, we will try to first identify those truncated objects and estimate the depth information for these objects based on other relevant points such as the conventional central point.
>
>
> ***L2: The method assumes that the ground is flat. However, the ground at the long distance and at the side is inconsistent with the nearby place. Therefore the virtual scene constructed from the infinite plane is not convinced.***
>
> Another limitation is that the proposed MoGDE is based on the implicit assumption that the ground plane is flat. For uneven ground planes such as slopes, continuing to use the flat ground depth estimation will lead to serious depth estimation errors. In Figure 9, we present an example of this situation where the ego vehicle is making a turn and sees a complex non-planar road surface with multiple vanishing points (VPs). In this case, adopting the flat ground depth estimation would not work properly. To solve this problem, one potential method is to segment the image from the bottom and check the consistency of identified VPs. If only one VP is detected, the ground is considered flat; otherwise, the ground depth should be separately estimated for each corresponding VP. We will work on this in our future work.
>
> ***L3: The formulas adopted in the equations are not unified. For example, the EL is suggested to be replaced by a single letter.***
>
> Thank you for pointing out the deficiencies in the formulas, we will unify them in the revised paper.

---

### Official Review · Reviewer_Nv1S · 2022-07-12

**Rating:** 6
**Confidence:** 4
**Soundness:** 4 excellent
**Presentation:** 3 good
**Contribution:** 3 good

**Summary:**

The main idea in the paper is to assume a flat ground plane and leverage known camera height above it etc to get depths for the ground points. The method refines the camera pose relative to this plane via horizon/vanishing point estimation in the images by following the DeepVP method[7] thus deriving depths for ground pixels.  For 3D detection, the idea is to fuse these depths with the image features using transformer layers, which pick suitable ground points for the object to derive its depth. For the final detector, the GUPNet head is used.

The results are claimed to be top on monocular 3D image detection on KITTI, which would be an impressive achievement. Suitable ablations for the method components are done. Additional results, showing the applicability of the introduced pose estimation components to detectors other than GUPNet are also presented.

**Questions:**

How exactly did you know where the vanishing point is on KITTI? I don't fully understand how you get the GT.

Please comment on flat plane / curvy road limitations of the method, if any.

Any way one can see which papers are monocular detection, on the KITTI leaderboard?



**Limitations:**

No specific limitations apart from what I mention in Questions above.

**Strengths And Weaknesses:**

Strengths:
- Insightful use of transformers to fuse estimated depths for object detection.
- Intuitive overall idea to leverage the flat ground plane assumption and estimate the camera pose relative to it -- it seems to make a difference in KITTI.
- Strong results on KITTI and meaningful ablations.

Weaknesses:
- The need for pose estimation on KITTI, what quantities were given and which were not, was not particularly clear. As a result, Sec 3.2 was a bit difficult to read. Consider clarifying this better in a subsequent version.
- The limitations of the method are not well explained. It assumes a flat ground plane which may not suit hilly scenes. Without reading the papers, it's not particularly clear to me how vanishing point estimation works when the road makes a turn. No experimental results showing model failures in such cases (or any cases really) are presented, in order to better understand the limitations / problems with the model.
- It is difficult to track based on the links presented, whether this method is #1 on KITTI for monocular depth or not. I am willing to believe it and I see MoGDE represented on the leaderboard here: http://www.cvlibs.net/datasets/kitti/eval_object.php?obj_benchmark=3d. It clearly outperforms GUPNet and MonoDETR, so perhaps the claim is correct.
- There are other CVPR 2022 methods on the leaderboard stronger than that are not necessarily cited (e.g. Diversity Matters: Fully Exploiting Depth Clues for Reliable Monocular 3D Object Detection, which has better numbers than MonoDETR).

---

> ### Author Response · Authors · 2022-08-02
> **Response to Reviewer Nv1S**
>
> ***W1: The need for pose estimation on KITTI, what quantities were given and which were not, was not particularly clear.***
>
> Our pose estimation network follows the framework of DeepVP [7], which is commonly used in visual odometry tasks. In Eq (1), the $f^{pose}$ is the CNN architecture used for horizon and vanishing point detection, we follow DeepVP and make modifications to the filters for the fully connected layers. The matrix $ \mathbf{A}$, formulated in Eq(9), represents the transfer matrix of the camera related to the motion of the ego-vehicle (pitch and roll angles). The $\mathbf{g}$ is a mapping function $\mathbf{g}:\left(\mathbb{R}^{2}, \mathbb{R}^{2}\right) \mapsto \mathbf{A}_{3\times 3}$ which turns pitch and roll angles into a matrix $\mathbf{A}$. During training, horizons and vanishing points are only used as intermediate results; they are not supervised with ground truth labels for training. The training label is the ego-vehicle motion transfer matrix after considering only pitch and roll angles (corresponding to matrix $\mathbf{A}$) provided by the KITTI Odometry dataset. In the revised paper, we will add more explanations about these quantities.
>
> ***W2: The limitations of the method are not well explained. It assumes a flat ground plane which may not suit hilly scenes. Without reading the papers, it's not particularly clear to me how vanishing point estimation works when the road makes a turn. No experimental results showing model failures in such cases (or any cases really) are presented, in order to better understand the limitations / problems with the model.***
>
> In L308 and Appendix C, we explain two main limitations of MoGDE. The first limitation is that the proposed MoGDE is based on the implicit assumption that the ground plane is flat. For uneven ground planes such as slopes, continuing to use the flat ground depth estimation will lead to serious depth estimation errors. In Figure 9, we present an example of this situation where the ego vehicle is making a turn and sees a complex non-planar road surface with multiple vanishing points (VPs). In this case, adopting the flat ground depth estimation would not work properly. To solve this problem, one potential method is to segment the image from the bottom and check the consistency of identified VPs. If only one VP is detected, the ground is considered flat; otherwise, the ground depth should be separately estimated for each corresponding VP. We will work on this in our future work.
>
> Second, the model heavily relies on the detection of the ground-contacting points. When these points are uncertain or invisible due to occlusion and truncation, our proposed ground-aware feature fusion method does not yield better results. To illustrate this limitation by visualization, we give the bad cases due to truncation in Figure 10 in the supplementary materials. In our future work, to tackle this limitation, we will try to first identify those truncated objects and estimate the depth information for these objects based on other relevant points such as the conventional central point.
>
> ***W3: It is difficult to track based on the links presented, whether this method is #1 on KITTI for monocular depth or not. I am willing to believe it and I see MoGDE represented on the leaderboard here: http://www.cvlibs.net/datasets/kitti/eval_object.php?obj_benchmark=3d. It clearly outperforms GUPNet and MonoDETR, so perhaps the claim is correct.***
>
> KITTI 3D object detection leaderboard does not differentiate between system models of the methods (e.g., LiDAR, binocular camera, monocular camera, etc.). In general, we can only identify those methods that are subordinate to Mono3D by the experimental settings given by the authors and the accompanying papers. The current rankings and detection accuracy results for monocular 3D object detection by monocular cameras are:
> | Rank |  Method  | Moderate |  Easy |  Hard |
> |:----:|:--------:|:--------:|:-----:|:-----:|
> |  316 |   MoGDE  |   17.88  | 27.07 | 15.66 |
> |  321 |  MonoDDE |   17.14  | 24.93 | 15.10 |
> |  327 |   DD3D   |   16.87  | 23.19 | 14.36 |
> |  331 | MonoDETR |   16.26  | 24.52 | 13.93 |
> |  343 |  MonoDTR |   15.39  | 21.99 | 12.73 |
> |  348 |  GUPNet  |   15.02  | 22.26 | 13.12 |
>
> ***W4: There are other CVPR 2022 methods on the leaderboard stronger than that are not necessarily cited (e.g. Diversity Matters: Fully Exploiting Depth Clues for Reliable Monocular 3D Object Detection, which has better numbers than MonoDETR).***
>
> This paper was not on the leaderboard at the time of submission and we will add this citation to the revised paper. This paper introduces a robust monocular 3D detector. It can generate multiple estimates of the depth for each target based on many different rules and assumptions, and combine the reliable estimates into a single depth to improve the reliability and accuracy of the depth detection.

---

### Official Review · Reviewer_nXFG · 2022-07-12

**Rating:** 7
**Confidence:** 4
**Soundness:** 3 good
**Presentation:** 4 excellent
**Contribution:** 3 good

**Summary:**

Targeting monocular 3d object detection, this paper exploits depth information by estimating ground plane and fuses that information with image features via attention layers. Experimental results on KITTI are very strong, outperforming MonoDETR by 2-4points.

**Questions:**

The paper is well written. For failure cases when ground-contacting points are hard to detect, does the method break fully or can it still output reasonable detections (i.e. it can learn to ignore those features)? including visualization of some failure cases would help better understand the effectiveness.

**Limitations:**

 As the authors mentioned, the proposed method is limited by ground-contacting points detection and could fail in cases where those points are occluded or truncated.

**Strengths And Weaknesses:**

Strength:
+ The proposed method is intuitive and clearly presented (i.e. it exploits depth information using ground plane). The paper provides a clear explanation and derivation of how the ground depth is estimated and used in the model.
+ Experiments are comprehensive and the results are strong. It outperforms previous sota by large margin (Tab.1), provides detailed ablation of different components (Tab.2), and gives good visualization of the output with attention maps.
+ Code is provided with high quality.

Weaknesses: the proposed method could be upbounded by ground-contacting point detection and assumes the ground plane is flat, while it works well on KITTI mono 3d detection, it could be less effective in general.

---

> ### Author Response · Authors · 2022-08-02
> **Response to Reviewer nXFG**
>
> ***W1: the proposed method could be upbounded by ground-contacting point detection and assumes the ground plane is flat, while it works well on KITTI mono 3d detection, it could be less effective in general.***
>
> In L308 and Appendix C, we explain two main limitations of MoGDE. First, the model heavily relies on the detection of the ground-contacting points. When these points are uncertain or invisible due to occlusion and truncation, our proposed ground-aware feature fusion method does not yield better results. To illustrate this limitation by visualization, we give the bad cases due to truncation in Figure 10 in the supplementary materials. In our future work, to tackle this limitation, we will try to first identify those truncated objects and estimate the depth information for these objects based on other relevant points such as the conventional central point.
>
> Another limitation is that the proposed MoGDE is based on the implicit assumption that the ground plane is flat. For uneven ground planes such as slopes, continuing to use the flat ground depth estimation will lead to serious depth estimation errors. In Figure 9, we present an example of this situation where the ego vehicle is making a turn and sees a complex non-planar road surface with multiple vanishing points (VPs). In this case, adopting the flat ground depth estimation would not work properly. To solve this problem, one potential method is to segment the image from the bottom and check the consistency of identified VPs. If only one VP is detected, the ground is considered flat; otherwise, the ground depth should be separately estimated for each corresponding VP. We will work on this in our future work.

---

### Meta-Review · Area_Chair_aBEr · 2022-08-26

**Recommendation:** Accept
**Confidence:** Certain

**Metareview:**

The paper received positive leaning reviews (2x borderline accept, 1x weak accept, 1x accept). The meta-reviewer agrees with the reviewers' assessment of the paper.

**Award:**

No

---

### Decision · Program_Chairs · 2022-09-14

Accept